# Retinal Responses to Visual Stimuli in Interphotoreceptor Retinoid Binding-Protein Knock-Out Mice

**DOI:** 10.3390/ijms241310655

**Published:** 2023-06-26

**Authors:** Marci L. DeRamus, Jessica V. Jasien, Jess M. Eppstein, Pravallika Koala, Timothy W. Kraft

**Affiliations:** Department of Optometry and Vision Science, University of Alabama at Birmingham, Birmingham, AL 35294, USA; jessicajasien@gmail.com (J.V.J.); jess.eppstein@gmail.com (J.M.E.); pkotla@plabs.com (P.K.)

**Keywords:** IRBP, retina, RPE, ERG, OKR, flicker, rod, cone

## Abstract

Interphotoreceptor retinoid-binding protein (IRBP) is an abundant glycoprotein in the subretinal space bound by the photoreceptor (PR) outer segments and the processes of the retinal pigmented epithelium (RPE). IRBP binds retinoids, including 11-cis-retinal and all-trans-retinol. In this study, visual function for demanding visual tasks was assessed in IRBP knock-out (KO) mice. Surprisingly, IRBP KO mice showed no differences in scotopic critical flicker frequency (CFF) compared to wildtype (WT). However, they did have lower photopic CFF than WT. IRBP KO mice had reduced scotopic and photopic acuity and contrast sensitivity compared to WT. IRBP KO mice had a significant reduction in outer nuclear layer (ONL) thickness, PR outer and inner segment, and full retinal thickness (FRT) compared to WT. There were fewer cones in IRBP KO mice. Overall, these results confirm substantial loss of rods and significant loss of cones within 30 days. Absence of IRBP resulted in cone circuit damage, reducing photopic flicker, contrast sensitivity, and spatial frequency sensitivity. The c-wave was reduced and accelerated in response to bright steps of light. This result also suggests altered retinal pigment epithelium activity. There appears to be a compensatory mechanism such as higher synaptic gain between PRs and bipolar cells since the loss of the b-wave did not linearly follow the loss of rods, or the a-wave. Scotopic CFF is normal despite thinning of ONL and reduced scotopic electroretinogram (ERG) in IRBP KO mice, suggesting either a redundancy or plasticity in circuits detecting (encoding) scotopic flicker at threshold even with substantial rod loss.

## 1. Introduction

Interphotoreceptor retinoid-binding protein (IRBP) is a large glycoprotein (~140 kDa) produced by the PRs and released into the interphotoreceptor matrix (IPM), where it is the most abundant soluble protein [1,2,3,4]. IRBP is restricted to the IPM by the tight junctions of the retinal pigment epithelium (RPE) and the junctional complexes of the outer limiting membrane [4,5,6]. IRBP is prescribed to mediate the transfer of retinoids, such as all-trans retinol and 11-cis retinal, between the retina and RPE [2,4,7,8,9,10,11,12,13,14,15,16,17] and contains three distinct retinoid binding sites [18]. Yet, the entirety of the signaling pathways impacted by retinoid trafficking between PRs, RPE cells, and Müller cells is unknown.

The transfer of retinoids between the RPE and retina is vital to the visual cycle [4,16]. The visual cycle starts in the rod and cone PRs, which contain 11-cis retinal, the light-sensitive chromophore of the visual pigments [19]. The photoisomerisation of 11-cis retinal leads to current and voltage changes and termination of the light-activated signal and includes release of all-trans retinal from the visual pigment and replacement with new 11-cis-retinal. Light capture photoisomerizes 11-cis retinal to the all-trans retinal conformation, which releases the enzymatic activity of the visual pigment to initiate the phototransduction cascade. Opsin bound to all-trans retinal is not light-sensitive, and thus, visual sensitivity is only restored when all-trans retinal is replaced with 11-cis retinal. All-trans retinal is recycled by one of two pathways, binding to IRBP in the IPM and then diffusing either to the RPE [16] or to Müller cells to feed into the newly described retinal visual cycle [20,21,22,23]. Trafficking of 11-cis retinal from RPE to the PR, via RPE, is also facilitated by IRBP [2,4,7,8,9,10,11,12,13,14,15,16,17]. Thus, evidence clearly indicates that IRBP plays an important role in the visual cycle.

Previous research on IRBP KO mice showed a reduction in the thickness of the outer nuclear layer (ONL), reduced scotopic and photopic electroretinogram (ERGs), and reduced photopic flicker amplitude [24,25,26,27,28,29,30]. Interestingly, one report showed a 30% decrease in the cone populations as measured by a primary antibody for mouse cone arrestin (mCarr) [24], a finding corroborated by other studies measuring peanut agglutinin (PNA), M-opsin, and S-opsin [25,30]. However, another report showed no difference in cone densities measured via PNA staining [28]. IRBP deletion has also been shown to reduce the amplitude and slow kinetics of cone responses without affecting cone adaptation in bright light conditions [31].

Here, we evaluate retinal structure with optical coherence tomography (OCT) and histological cone density measurements. We corroborated structural tests by evaluating functional visual responses to complex stimuli, as well as behavioral tasks, in mice lacking IRBP at 4–6 weeks old. We examined WT and IRBP Heterozygous (Het) mice at 4–6 weeks and 13–15 months old. Thus, we tested spatial frequency acuity, contrast sensitivity, photopic and scotopic critical flicker frequency (CFF) testing, as well as measures of RPE function via ERG C-wave analysis, and inner retina function via oscillatory potentials (OPs). Finally, the role of IRBP in a recently described adaptive potentiation (AP) was examined, and results were consistent with an important role of IRBP in AP as well.

## 2. Results

### 2.1. Optical Coherence Tomography (OCT)

OCT of IRBP KO mice showed a reduction in PR layer thickness compared to wildtype (WT) and IRBP Het mice. OCT was used to measure the ONL, PR, inner nuclear layer (INL), and full retinal thickness (FRT) in 1-month-old mice (Figure 1). The ONL of the IRBP KO mouse (46 ± 1 µm) was almost 30% thinner than WT (65 ± 2 µm) and IRBP Het mice (64 ± 1 µm). In addition, IRBP KO mice (28 ± 1 µm) showed 35% thinner PR thicknesses than WT (43 ± 3 µm) and IRBP Het mice (39 ± 3 µm). There were no differences in average INL thickness between WT (28 ± 2 µm), IRBP Het (28 ± 2 µm), or IRBP KO mice (27 ± 3 µm). Finally, IRBP KO mice (182 ± 2 µm) also had 20% thinner FRT than WT (227 ± 5 µm) and IRBP Het mice (230 ± 4 µm).

OCT was also performed in WT and IRBP Het mice at 13–15 months of age to determine whether a slow PR degeneration occurred in IRBP Het mice. FRT was reduced by 18% in older IRBP Het (167 ± 9 µm, *p* < 0.05) compared to WT mice (203 ± 5 µm). Older IRBP Het mice showed a 35% reduction in PR thickness (26 ± 2 µm) compared to WT mice (40 ± 8 µm). There were no differences in ONL or INL thicknesses between IRBP Het and WT mice at the tested older age. Retinal degeneration in IRBP KO mice was too significant to measure layer thickness at 13–15 months.

### 2.2. Immunohistochemistry for mCarr Assessment of Cone Density

Cone PR cells were counted via staining with antibodies to cone arrestin; mCarr positive cells were counted across a single vertical section of the retina that included the optic nerve. The retina was divided into 8 regions, four superior and four inferior to the optic nerve (Figure 2). The average of these 8 regions showed that IRBP KO mice had 15% fewer mCarr positive cell bodies (119.3 ± 3.4/mm) than WT mice (140.3 ± 7.5/mm) across the retina (*n* = 4 for each KO and WT, *p* = 0.02). The losses in the superior peripheral retina were not different from losses in the inferior peripheral retina, suggesting similar losses in S and M cones.

### 2.3. Electroretinogram (ERG)

The IRBP KO mice responded to light flashes with smaller and slower a- and b-waves (Table 1, Figure 3A,B) under both scotopic and photopic conditions. IRBP Het animals showed no ERG response differences compared to WT.

Under dark-adapted conditions, WT and IRBP Het mice had higher a- and b-wave amplitudes, higher a/b wave ratios, faster b-wave implicit time (ttp), higher half-saturating intensities (I_½_), and larger oscillatory potential magnitudes (OP areas) than IRBP KO mice (Figure 3). Maximum a-wave response amplitudes to a bright saturating flash were two-fold reduced in IRBP KO mice (195 ± 17 µV) compared to WT (434 ± 39 µV) and IRBP Het mice (429 ± 41 µV). Maximum b-wave response amplitudes to a bright saturating flash were also reduced in IRBP KO mice (616 ± 58 µV) compared to WT mice (819 ± 58 µV), but the b-wave loss was only half of that of the a-wave reduction, suggesting some synaptic compensation mechanism between PRs and bipolar cells.

Consequently, IRBP KO mice had lower a/b wave ratios (32 ± 1%) compared to WT (52 ± 2%) and IRBP Het mice (52 ± 2%). The b-wave implicit time for IRBP KO mice (93 ± 3 ms) was significantly slower than WT (70 ± 3 ms) and IRBP Het mice (73 ± 3 ms). The a-wave implicit time in IRBP KO mice (13 ± 1 ms) was 39% slower than IRBP Het mice (8 ± 0.3 ms).

To fully characterize the range of retinal function, the ERG responses were recorded at 2 ms flashes, producing from 0.14 (± 0.01) to 1698 (± 139) R* per rod. The dimmest rod-driven response amplitudes (Figure 3C, scotopic threshold response, 0.14 ± 0.01 R* per rod), were 51% lower in IRBP KO mice (19.9 ± 1.3 µV) compared to WT (40.5 ± 4.5 µV) mice. The intensity at which a half-maximal b-wave response was evoked. (I_½_) was calculated by plotting the response amplitude by the log of the intensity for each of the intensities tested (Figure 3D). The I_½_ for IRBP KO b-wave (15.6 R* per rod) mice was not significantly lower than WT (21.9 R* per rod) and IRBP Het mice (13.8 R* per rod, Table 1).

Under photopic conditions, WT and IRBP Het mice also had larger a-wave amplitudes and a/b ratios than IRBP KO mice (Table 1 Light-adapted). Light-adapted IRBP KO mice showed a significant reduction in a-wave amplitude (109 ± 13 µV vs. 252 ± 31 µV for WT, *p* < 0.001); but the light-adapted a-wave implicit times were not different between the two strains of mice. WT mice had 39% larger photopic b-wave amplitudes (384 ± 46 µV, *p* < 0.05) than IRBP KO mice (233 ± 28 µV). The photopic a- to b-wave ratios for IRBP KO mice (49 ± 1%) were lower than WT (68 ± 4%, *p* < 0.001) and IRBP Het mice (66 ± 2%, *p* < 0.05).

To look at inner retinal function, the area of the OP (Figure 3E,F), isolated from the rising phase of the dark-adapted b-wave, was measured and found to be 44% smaller in IRBP KO mice (2.2 ± 0.2 µV*ms) compared to WT (3.9 ± 0.2 µV*ms) and IRBP Het mice (3.9 ± 0.2 µV*ms). OP area, under photopic conditions, was 1.9 ± 0.2 µV*ms for IRBP KO mice, which is more than 17% lower than both WT (2.6 ± 0.2 µV*ms) and IRBP Het mice (2.3 ± 0.2 µV*ms, *p* < 0.05).

To determine if IRBP plays a key role in adaptive potentiation (AP) previously reported by our lab [32,33], massed PR response recordings were performed in isolated tissues with synaptic blockers. In WT retinas, the average response to a saturating test flash was 114 ± 16 µV, and it was more than doubled by adaptive potentiation (240 ± 53 µV, *n* = 9 trials, *n* = 4 animals), following a 3 min conditioning light exposure (Figure 4). The isolated retinas from IRBP KO mice had significantly smaller saturating response amplitudes at the outset (37.1 ± 3.8 µV) as well as reduced overall sensitivity in their isolated tissue response. Further, the adaptive potentiation in the IRBP KO retinas showed only a 27.6 ± 5.9% potentiation (47.3 ± 5.3 µV; *n* = 11 trials, *n* = 3 animals), a four-fold reduction in the level of AP compared to WT retinas. Retinas from IRBP Het mice showed a slight reduction in the saturating response (84.6 ± 9.6 µV) and 76.7 ± 12.3% potentiation (161 ± 29 µV; *n* = 17 trials, *n* = 6 animals) that was not statistically different from the measurements from WT retinas.

As a measure of the slow processes of potassium removal by RPE and Müller cells, the c-wave responses to 5 s step of dim or bright light (1530 and 42,300 R* per rod/s) were also recorded (Figure 5). At both intensities, there were no differences in the c-wave amplitudes generated: dim step responses of 242 ± 32 µV for WT (*n* = 13) vs. 266 ± 34 µV (*n* = 10; *p* = 0.62) for IRBP KO; bright step responses of 365 ± 38 µV for WT vs. 313 ± 37 µV for IRBP KO (*p* = 0.36). There were also no differences in the time constant of the c-wave, tau, at low intensity (WT: 1.40 ± 0.10 s, KO: 1.30 ± 0.23 s; *p* = 0.43) or high intensity.

Scotopic and photopic CFF were measured with a new flicker sweep ERG technique. Both scotopic and photopic flicker responses showed reduced amplitude in IRBP KO mice, as expected from the b-wave flash responses (Figure 1). The IRBP KO mice had 19% lower photopic CFF (33 ± 2 Hz) than WT (41 ± 2 Hz) and IRBP Het mice (41 ± 3 Hz). However, the scotopic CFF was similar (22 Hz) in all genotypes, despite loss of rod PRs in the IRBP KO mice (Figure 6).

### 2.4. Optokinetic Reflex (OKR)

IRBP KO mice have reduced scotopic and photopic contrast sensitivity compared to WT and IRBP Het mice (Figure 7A,B). Under scotopic and photopic conditions, there were no differences in contrast sensitivity between WT and IRBP Het mice. One-way ANOVA showed a significant difference in both photopic and scotopic contrast sensitivity (*p* < 0.05) between the knockout and other groups.

IRBP KO mice had reduced spatial frequency acuity compared to WT and IRBP Het mice. Under both scotopic and photopic conditions, WT and IRBP Het mice showed higher spatial frequency acuity than IRBP KO mice (Figure 7C). IRBP KO mice had more than 12% reduction in scotopic frequency acuity (0.261 ± 0.002 c/d, *p* < 0.001) compared to WT (0.298 ± 0.003 c/d) and IRBP Het mice (0.301 ± 0.004 c/d). Photopic frequency acuity was also reduced by more than 9% in IRBP KO mice (0.276 ± 0.002 c/d, *p* < 0.001) compared to WT (0.314 ± 0.002 c/d) and IRBP Het mice (0.306 ± 0.003 c/d). However, under photopic conditions, IRBP Het mice frequency acuity was also lower than WT mice (*p* = 0.009).

## 3. Discussion

Our results indicate that both scotopic and photopic ERG responses are reduced in one-month-old adult mice when IRBP is absent. These general findings are in agreement with the original descriptions of this line of mice [24,25,26,34]. While the loss of IRBP resulted in a two-fold reduction of the a-wave (55% scotopic and 57% photopic), the b-wave was only reduced by a quarter to a third (25% scotopic and 39% photopic). These results suggest higher synaptic gain between PRs and bipolar cells in retinas lacking IRBP, which if present could indicate a compensatory mechanism in response to damaged/missing rods. The dimmest light response measured was of a stimulus strength of well below a single photoisomerization per rod per flash (0.14 R*) and defined the threshold responses of our experiments. We extrapolate the linearity of this response to indicate a reduced scotopic threshold response (pSTR; [35]), which implies that rod loss is more critical to pSTR than CFF.

The OPs are generated by bright light flashes that activate both rod- and cone-driven circuitry under dark-adapted (DA) conditions, and when the retina is light-adapted (LA), they show only cone-driven responses. Thus, the finding that DA and LA OPs are reduced is consistent with reduced rod-driven dim and bright light response but, additionally, points to loss of cone-driven IPL responses. The OP area was reduced by 44% under dark-adapted conditions and by 17% under light-adapted conditions, suggesting a significant loss in cone-driven signals in bipolar cell to amacrine and amacrine to ganglion cell signaling in the IPL. The relative preservation in light-adapted responses could be explained by summation differences in receptive fields with small numbers of cones. We speculate that dark-adapted signals originate in larger receptive fields that require complete cone mosaic integrity, whereas light-adapted smaller receptive field signals are less affected by the 15% cone loss and more of those signaling circuits are intact.

The I_½_, the intensity at which the b-wave response reaches half of its maximum, was unchanged in IRBP KO mice, indicating that despite the loss of PRs the operating range of the retinal bipolar cells was unmoved. That result is consistent with an increase in gain of the rod-to-bipolar signaling (synapse). The a- and b-wave peak responses occurred later in dark-adapted IRBP KO mice, but the photopic ERG timing was unchanged. In contrast, the scotopic, rod-based CFF was unchanged in IRBP KO mice, but the photopic, cone-based CFF was significantly reduced in IRBP KO mice. These results suggest that the cone circuits are more sensitive to PR loss and agree with a similar finding in light-damaged albino rats with modest losses of cone PRs [36,37].

A general loss of PRs was observed in IRBP KO mice and demonstrated by the thinning of the ONL and PR. Previous reports of cone density in IRBP KO mice were conflicting [24,28]. Our measures of cone density in IRBP KO mice confirmed a reduction in cones, indicating that some of the ONL thinning is due to cone loss, but at a value of 15%, half of that previously reported by Parker et al. [28]. Reductions in photopic ERG responses were also consistent with loss of cones. However, the small percentage of cones in the murine retina means that rod loss is primarily responsible for the measured ONL thinning. Functionally, scotopic flicker sensitivity is normal in IRBP KO mice despite thinning ONL and reduced scotopic ERG, suggesting either a redundancy or plasticity in detecting scotopic flicker at threshold even in the face of substantial rod loss. Consistent with our ERG results, behavioral abnormalities in IRBP KO mice indicate that the absence of IRBP reduces the ability of the retina to respond to demanding visual stimuli such as high to low contrast or high spatial frequency.

AP is a newly described form of adaptation in which a period of rod-saturating light results in a short-lived but dramatic increase in the dark circulating current of rods and enhances human scotopic sensitivity along an identical time course [32,33]. Exogenously supplied all-trans-retinol or high concentrations of all-transretinal affect PR sensitivity and are involved in AP. All-trans retinol is thought to be part of a feedback loop, resulting in AP by increasing sensitivity of rods [32,33]. IRBPs role in transfer of retinoids coupled with retinoids’ role in AP implies that IRBP could be involved in this form of adaptation. Our experiments attempted to optimize the stability of tissue and minimize turbulence that could disrupt the IPM surrounding and the PR surface of the isolated retina. In fact, the magnitude of the AP found in WT mice (111%) in the experimental conditions reported here surpassed that reported in our earlier publications. While AP was present in IRBP KO mice, it was almost 4-fold smaller than that found in WT mice, supporting the idea that IRBP plays an important role in the mechanism of AP. However, AP was not blocked completely, suggesting that all-trans retinal released into the IPM can partially activate the feedback loop in the absence of IRPB as a binding partner. Future research directions might include searching for alternative binding partners for retinoids, but multiple non-specific transporters may be present in the interphotoreceptor matrix.

The c-wave is derived from the removal of a potassium load in the outer retina by a combination of mechanisms acting via the RPE and Müller cells. The significant shift in the ratio of the dim vs. bright light-activated c-waves IRBP KO animals may be due to the relative losses of rods versus cones in the outer retina and is consistent with the shift in the rod-to-cone ratio from about 30:1 to 19:1 in these retinas.

## 4. Materials and Methods

### 4.1. Experiments with Animals

All animal procedures performed in this study were approved by the Institutional Animal Care and Use Committee at the University of Alabama at Birmingham. All animals were maintained on a standard 12/12 h light cycle. IRBP Het mice were obtained from Jackson Laboratories (Bar Harbor, ME, USA), and bred to produce WT, IRBP Het, and IRBP KO genotypes. Mice of either sex at 30 ± 2 days postnatal were used for all experiments, except where noted. Genotyping was performed as suggested by Jackson labs protocol Rbp3^tm1Gil^-Alternate 1. Genomic DNA extraction and PCR amplification was performed using the KAPA Biosystems standard mouse genotyping kit (KK7352). PCR was performed using the following cycling parameters: 94 °C denaturation for two minutes, for one cycle; ten cycles of 94 °C for twenty seconds, 65 °C for fifteen seconds, 68 °C for ten seconds; 28 cycles of 94 °C for fifteen seconds, 60 °C for fifteen seconds, 72 °C for ten seconds; and one cycle of 72 °C for two minutes. Primer sequences used were (12834 Jackson, mutant sense) GCT ACT TCC ATT TGT CAC GTC C, (19061 Jackson, wildtype sense) GGA CCC ACA CCT GAA GAC AG, and (19062 Jackson, common anti-sense) CAT ATC CAC ACC TGC CAA CA. PCR products were 147 bp for WT, 350 bp for IRBP KO, and both a 147 and 350 bp band for IRBP Het mice. PCR products were run on a 1% agarose gel (Thermo Fisher Scientific; Waltham, MA, USA) at 100 V for 30 min. Genotypes were confirmed with standard Sanger sequencing methods.

Animals (*n* = 8) were also tested for the RPE65 mutation using PCR restriction fragment length polymorphism method. DNA from mouse tails was extracted via incubation in 0.05 M NaOH at 95 °C and 500 rpm for 20 min and then amplified using the forward 5′-ACCAGAAATTTGGAGGGAAAC-3′ and reverse 5′-CCCTTCCATTCAGAGCTTCA-3′ primers [38]. One µg of the product (545 bp) was digested for 36 h using 50 units of a *MwoI* restriction enzyme (New England Biolabs, Ipswich, MA, USA) and run on a 3% agarose gel. Digestion of the product would produce fragments of 180 and 365 bp to indicate a standard Met-450 genotype. The Leu-450 variant eliminates the *MwoI* site and is unaffected by the digestion and produces the same 545 bp band. All animals tested showed the standard Met-450 genotype.

### 4.2. Optical Coherence Tomography

ONL, INL, PR, and FRT were measured using OCT (Bioptigen 840 nm; Phoenix-Micron, Inc., Bend, OR, USA). OCT images were obtained in WT, IRBP Het, and IRBP KO mice at 1 month postnatal. OCT collection and layer thickness analyses were performed using previously published methods [39], and 10 marks corresponding to specific reflective layers of the retina were placed using Bioptigen Diver 2.4 software. FRT was measured as the difference between marker 1 and 10 (Figure 1), corresponding to the inner limiting membrane and the outer edge of the RPE, respectively. The ONL was measured between markers 5 and 6, corresponding to the hyporeflective region above band 1. The INL was measured between markers 3 and 4, corresponding to the hyporeflective region above the ONL. Finally, the PR was measured between markers 6 and 9, corresponding to bands 2 and 3, which encompasses both the inner and outer segments, respectively. Measurements were determined at eight equidistant eccentricities in the nasal and temporal directions from the optic nerve head (±0.56, ±0.42, ±0.28, ±0.14). The average ONL, INL, PR, and FRT for all eccentricities were calculated and two-tailed *t*-tests were performed to determine differences between genotypes.

### 4.3. Histology

Following the protocol of Benthal, McKeown & Kraft, Hertfordshire, UK, (2022), eyes were fixed in 4% paraformaldehyde for 4 h at room temperature, washed in 1X PBS, and then placed in 30% sucrose overnight at 4 °C. Retinas were cryosectioned into 10 μm slices onto superfrost plus slides and left to dry for 1 h at 37 °C. Slides were stored at 4 °C prior to immunohistochemistry. Slides were removed from the freezer and placed in acetone for 3 min, followed by one hour incubation at 50 °C. Blocking was performed for 1 h at room temperature using 20% goat serum (Jackson ImmunoResearch Laboratories, West Grove, PA, USA) in 1X PBS. Slides were incubated in mCarr (LUMIj [40], 1:500; kindly provided by Dr. Cheryl M. Craft, University of Southern California, for 1 h at room temperature, followed by three washes for 5 min each in 2% goat serum in 1X PBS. Slides were incubated in a secondary antibody with an Alexa-488 fluorophore (1:200, Sigma, St. Louis, MO, USA) in a light-tight box for 1 h at room temperature, again followed by washes in 2% goat serum in 1X PBS. Slides were incubated in DAPI (ThermoFisher Scientific; Waltham, MA, USA) for 10 min at room temperature. Slowfade gold antifade mountant (ThermoFisher Scientific, Waltham, MA, USA) was used to coverslip. Stained tissues were stored at 4 °C prior to viewing. Images used for cell counting were obtained on a microscope at 40× magnification (BX51; Olympus, Tokyo, Japan). Cone cell counts were determined by counting the number of mCarr positive cone pedicles within four 500 µm regions superior and inferior to the optic nerve. Cone pedicles that were close or overlapping were distinguished by counting the number of outer segments that were visible in the appropriate region. Pedicles that were dim or out of focus and lacked outer segments were excluded. In all other cases, pedicles were counted whether their outer segment was visible or not. Three independent observers performed cone cell counts, blind to genotype, and an average of the three counts were used in analyses. Independent two-tailed *t*-tests were performed to determine group differences in the number of mCarr positive cones per mm, across the entire retina.

### 4.4. Electroretinogram

The mice were prepared for ERG as previously performed and described [41,42]. Mice were dark adapted for 1 to 3 h and then anesthetized with ketamine/xylazine (IVX Animal Health, 90.9 µg/g; Lloyd Laboratories, 9 µg/g). Corneas were anesthetized with 0.5% proparacaine (Bausch & Lomb, Vaughan, ON, Canada) and pupils dilated with 1% tropicamide (Alcon Laboratories, Geneva, Switzerland) and 2.5% phenylephrine (Ocusoft Inc, Rosenberg, TX, USA). Body temperature was maintained by a heating pad (Braintree Scientific, Braintree, MA, USA), and 2.5% methylcellulose (Goniosol; Ciba Vision, Batam City, Indonesia) was applied to the corneal surface. A platinum wire electrode (2 µm diameter) was placed on the tip of a fiber optic cable, which delivered a light stimulus from a 100 W halogen bulb (Xenophot HLX 64623; Osram, Quezon City, Philippines) controlled by a constant power source (ATE 15-15 M; Kepco Power Supplies, Flushing, NY, USA). The stimuli were typically delivered to the left eye, and a reference electrode consisting of a gold loop was placed touching the cornea of the right eye. Several ERG testing parameters were employed. First responses to 2 ms, 505 nm (bandwidth 35 nm) flashes of increasing intensity (0.14 ± 0.01 to 1700 ± 140 R* per rod) were measured. The intensity was attenuated using neutral density filters coated with metallic spray, Inconcel, to reduce transmission by calibrated amount of approximately 0.3, 0.6 and, 1.2 log units. In order to calculate R*, a collecting area of a rod at the pupil of 0.11 µm^2^ was assumed based on previously described calculations [43,44]. Repeated responses ranging in number from 3–30 were averaged, depending on the signal-to-noise ratio. Interstimulus interval ranged from 2–90 s, with shorter intervals for dimmer intensity stimuli. A saturating response was elicited using a camera flash unit which delivered a 2 ms, 500 nm flash (10 nm bandwidth) of about 166,000 R* per rod. To isolate cone responses, the same bright 500 nm flash was presented on a background light of 505 nm (2020 ± 160 R*/rod-s), following 3 min of adaptation.

The OPs form a complex wave, separated from the standard ERG by band pass filtering (34 to 70 Hz). The OPs have contributions from amacrine and ganglion cells with some input from the a-wave when the stimuli are bright enough to generate a fast a-wave. In order to quantify the OP signals, OP area was measured between 0.015 and 0.15 ms for each individual trial, and then, the group results were analyzed by a two-tailed *t*-test [45,46]. The area of the recording noise for an equal time period was subtracted from the individual OP measures. Retinoids are transferred to the RPE, ostensibly via IRBP, so it is possible that RPE function would be impaired in the absence of IRBP. To measure RPE function, c-wave responses were recorded in response to 5 s steps of light (1530 and 42,300 R*/rod-s), presented while recording responses in a DC configuration [47]. The c-wave stimuli were delivered via fiber optic, but the recording electrodes for this later set of studies was a mouse contact lens with embedded gold wire (LKC Inc., Delhi, India); thus, the R* values may be overestimated because of reflections off of the contact lens.

The CFF was also determined following the protocol of DeRamus and Kraft [48]. To evoke the flicker response, an LED stimulus was pulse-width modulated over 5 s to produce a linear sweep of frequencies in a single trial; the range was 0.1–30 Hz for scotopic and 0.1–55 Hz for photopic conditions. A high efficient luminance green (525 nm, bandwidth 36 nm) LED light source (LZ1-00G102, LED Engin, San Jose, CA, USA) with an unattenuated mean power of 7.42 W/s was used. Under scotopic conditions, the LED was attenuated by 3.49 log units using Wratten neutral density filters, resulting in a mean power of 2.4 mW/s. A 94% Michelson’s contrast was produced with a minimum and maximum intensity of 0.468 and 14.56 W/s, respectively. As a baseline measure, responses to a steady step of light of equal luminance were recorded before and after the flickering stimuli. A fast Fourier transform (FFT) of the baseline and flickering responses were calculated, and the difference between the two was determined to create a difference spectra (Appendix A). The CFF value was determined by fitting a line to the log of the difference spectra and calculating the frequency at 6.18 log µV^2^ response, which was previously determined to be equivalent to a 3 µV threshold used to determine CFF using the traditional steadily flickering stimuli [48,49].

### 4.5. Isolated Retina Electroretinogram

Isolated retina ERG was recorded as previously described [32,47]. Briefly, mice were dark adapted overnight and eyes enucleated under infrared illumination. The retinas were isolated under a dissection microscope (MS-5; Leica) in HEPES Buffer that contained (mM): 140 NaCl, 3.6 KCl, 2.4 MgCl_2_, 1.2 CaCl_2_, 3.0 HEPES, 10 BaCl_2_, and 10 Glucose. The retinas were mounted PR side up, with a filter blocking all but a central 2 mm disk of retina. The retinas were perfused under Bicarbonate Locke’s Buffer that contained (mM): 120 NaCl, 20 NaHCO_3_, 3.6 KCl, 2.4 MgCl_2_, 1.2 CaCl_2_, 3.0 HEPES, 0.02 EDTA, 1.14 mL AP-4, 0.33 g D/L aspartate, and 10 Glucose. Perfusion flow rate was 0.5 mL/min maintained at 35 °C and a pH of 7.5. Responses to 10 and 200 ms steps of increasing intensities were recorded until a saturating response was obtained for both stimulus durations. The retina was then exposed to 3 min of the rod-saturating light. Following this 3 min conditioning light exposure, 10 ms saturating flashes were presented to measure the saturating rod circulating current. Responses below 20 µV and with more than 20% baseline drift were excluded from analyses. Recordings were made in a DC configuration, with a 300 Hz low-pass filter.

### 4.6. Optokinetic Reflex

The OKR is a slow reflex-like head movement that can be evoked in response to rotating contrasting bars in a virtual drum [50,51]. The OKR was evoked using the OptoMotry system (Cerebral Mechanics) [52,53,54], which consists of four computer monitors arranged to form a virtual drum around an arena with a raised central platform. A sine wave grating is displayed by the computer monitors, and the center of the virtual drum is continually adjusted by the operator, via a live video, to reflect the animal’s viewing position by the placement of a cursor on the mouse’s head. The grating moved at a constant speed of 12 s/d for all tests. Both acuity and contrast sensitivity threshold were determined via a stairstep protocol, with smaller steps each time a reflex was not observed [55]. Spatial frequency acuity was determined by keeping the contrast of the bars constant at 100%, while slowly increasing the spatial frequency, starting at 0.05 c/d and stopping at the point at which the reflex was no longer evoked. Contrast sensitivity threshold was determined by maintaining a fixed spatial frequency and slowly decreasing the contrast of the bars, starting at 100% and stopping at the point at which the reflex was no longer evoked. Contrast sensitivity threshold was determined at six different spatial frequencies (0.031, 0.064, 0.092, 0.103, 0.192, and 0.272 c/d). Our system was adapted to make scotopic measurements by adding calibrated neutral density filters (4.64 to 4.72 ND) over the computer monitors to yield a very low, matched luminance for the four monitors that make up the virtual drum. The OptoMotry was adapted to include an internal infrared light source (Osram Infrared Emitters-High Power Infrared 850 nm) to allow visualization of the animal with the equipped infrared sensitive camera.

Luminance of the computer monitors, obtained via a J-16 Digital Photometer (Spectronics; Westbury, NY, USA), was calculated using the Michelson contrast (maximum − minimum)/(maximum + minimum). Luminance measures were obtained in eight directions around the compass for the black (minimum) and white (maximum) screens. The black screen had an average luminance of 0.0973 ± 0.0015 foot-candles; the white screen had an average luminance of 23.7 ± 0.8 foot-candles. The Michelson contrast for the OptoMotry system was 99.18%. The inverse of the contrast threshold obtained on the OptoMotry was corrected by the calculated Michelson’s contrast for the computer monitors.

Spatial frequency acuity group differences were tested using a two-tailed *t*-test and contrast sensitivity group differences tested using a one-way ANOVA with post hoc two-tailed *t*-test to account for type 1 errors (false positives).

All animal experiments and protocols were reviewed and approved by the Institutions Animal Care and Use Committee, protocol #09617.

## 5. Conclusions

Loss of IRBP results in significant structural changes within the retina, affecting both rod and cone function by 1 month of age. Reduced OPs in dark-adapted conditions derive from both rod- and cone-driven circuits as bipolar cells interact with amacrine and ganglion cells. The loss of rods in the ONL must weaken the driving signals participating in the DA OP generation and STR results. The reduction of the ONL by over 30% in OCT shows the dramatic effect on rod loss, and the 15% loss of cones contributes minimally to the ONL thinning because of the low percentage of cone photoreceptors in the mouse retina. The motion detection circuits, as represented by OKR tests, show about equal loss in scotopic and photopic vision, despite the much greater loss of rod photoreceptors. This result may indicate a greater reserve or redundancy in rod-driven circuits compared to those driven by cones. Flicker detection shows only a loss of cone-driven signals. Again demonstrating that the cone circuits for flicker are more delicate or exhibit less redundancy or plasticity than rod-driven circuits.

## Figures and Tables

**Figure 1 ijms-24-10655-f001:**
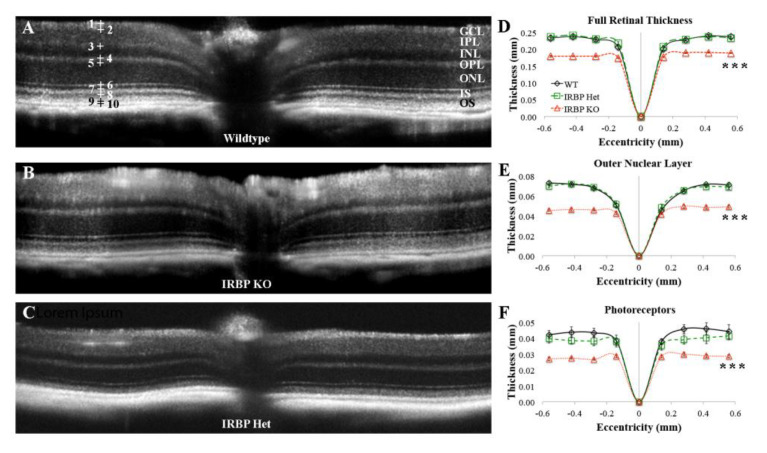
Optical coherence tomography analysis. OCT images for representative 4–6 week-old WT (**A**), IRBP KO (**B**), and IRBP Het (**C**) mice are shown; thickness scales for OCT are represented in (**D**). Ten marks (+) identify specific layers or layer-boundaries on the retina. OCT analysis of 4–6 week-old mice showed that IRBP KO mice had reductions in (**D**) FRT, (**E**) ONL, and (**F**) PR thicknesses compared to WT and IRBP Het. There were no differences between any of the groups for INL thickness measured at 4–6 weeks. Note: Different y scales for **D**–**F**. (*** *p* < 0.001) (4–6 wks: WT *n* = 5, IRBP KO *n* = 9, IRBP Het *n* = 8; 13–15 months: WT *n* = 6, IRBP Het *n* = 4).

**Figure 2 ijms-24-10655-f002:**
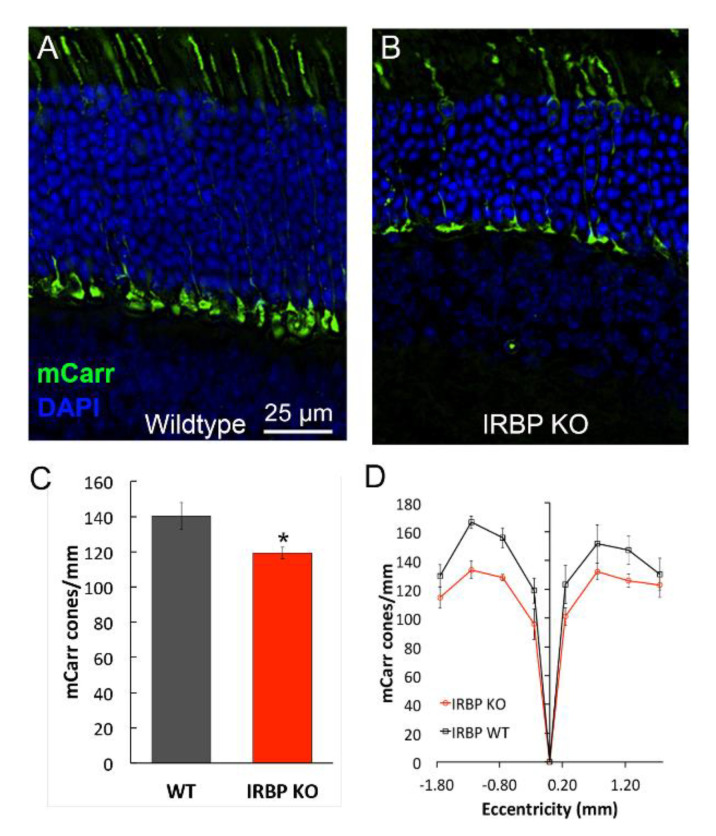
Measures of cone numbers via mCarr immunohistochemistry at 4–6 weeks. (**A**) WT and (**B**) IRBP KO mCarr immunohistochemistry images. (**C**) mCarr positive cones per mm across the entire retina for WT and IRBP KO mice. (**D**) mCarr positive cones per mm in WT and IRBP KO mice at eight different regions on either side of the optic nerve (nasal and temporal directions). Green = mCarr, Blue = DAPI, scale bar for A, and B = 25 µm. (* *p* < 0.05) (WT *n* = 4, IRBP KO *n* = 4).

**Figure 3 ijms-24-10655-f003:**
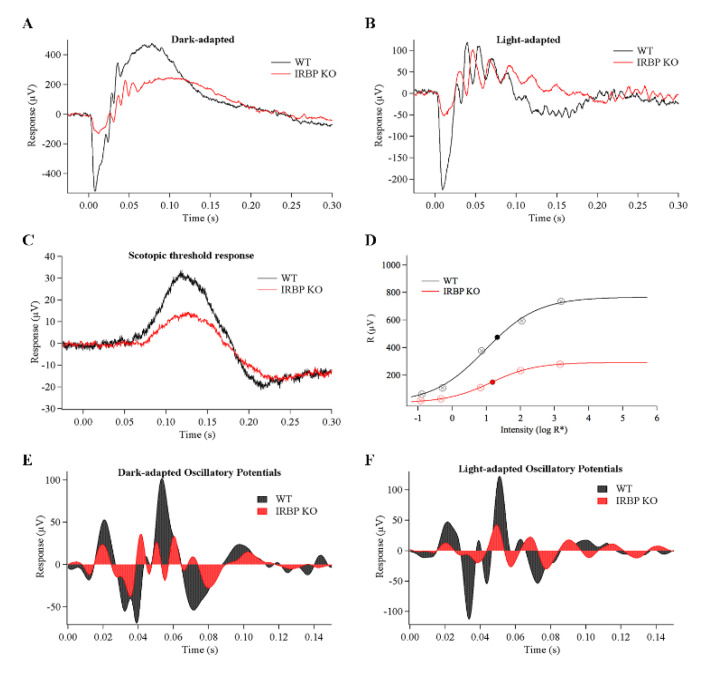
Electroretinogram ERG recordings of 1-month-old WT and IRBP KO mice. (**A**) Dark-adapted bright flash responses (166,000 R* per rod) and (**B**) light-adapted responses to a flash of the same intensity (background light 505 nm, 2020 ± 160 R*/rod-s) showed reductions in IRBP KO mice. Average of three responses, AC bandpass filter 1.6–300 Hz. (**C**) Average scotopic threshold responses (0.14 ± 0.01 R* per rod) show reductions in IRBP KO mice. Average of 20 responses, AC bandpass filter 1.6–300 Hz. (**D**) Intensity response curves show reduced maximum amplitudes for b-waves of the IRBP KO but no difference in the I ½ compared to WT (black- and red-filled circles). (**E**) Typical recordings of the dark-adapted and (**F**) light-adapted oscillatory potentials, isolated from the standard ERG response by band pass filtering (34 to 70 Hz). OP area, demonstrated by the fill-to-zero coloring was measured between 0.015 and 0.15 ms. OP area was reduced in IRBP KO mice compared to WT (Table 1). All mice were 30 ± 2 days old. (WT *n* = 12, IRBP KO *n* = 11, IRBP Het *n* = 7).

**Figure 4 ijms-24-10655-f004:**
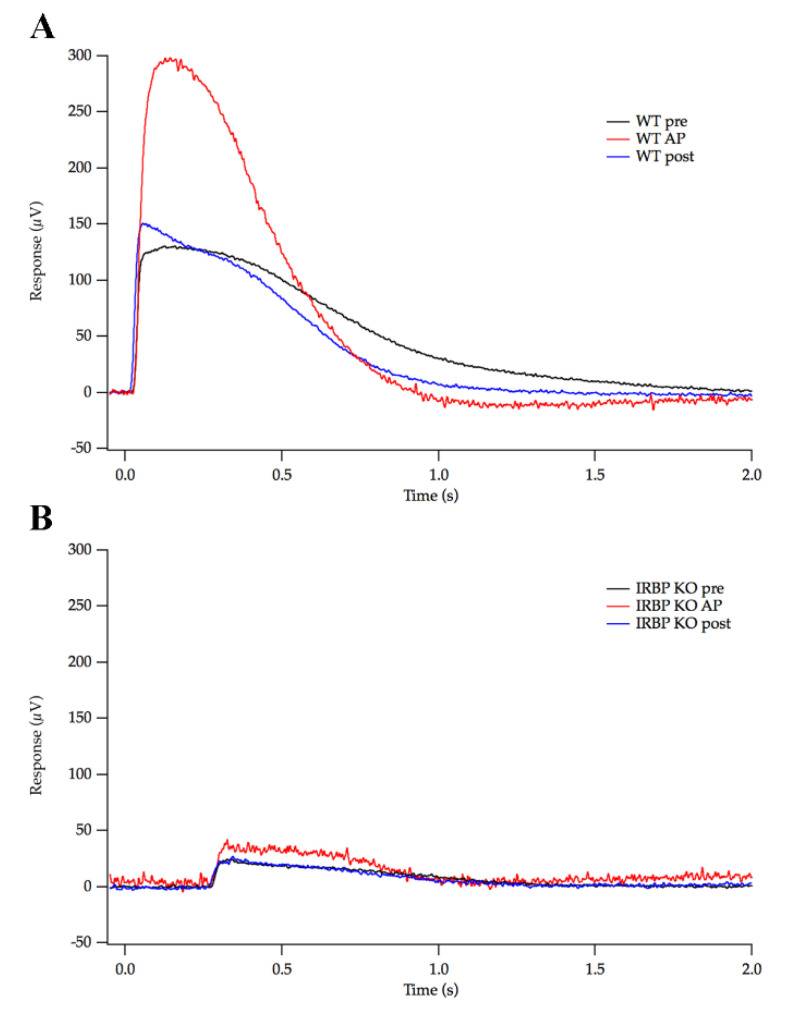
PR responses measured via isolated retina ERG. (**A**) Representative WT responses to a saturating test flash before (pre), 5 s after (AP), and 215 s (post) after a 3 min conditioning step of light. WT AP responses were double that of the pre response. (**B**) Representative IRBP KO responses that were significantly lower than WT and showed a four-fold reduction in the level of AP compared to WT. (WT *n* = 9 trials and *n* = 4 mice, IRBP KO *n* = 8 trials and *n* = 3 mice, IRBP Het *n* = 17 trials and *n* = 6 mice).

**Figure 5 ijms-24-10655-f005:**
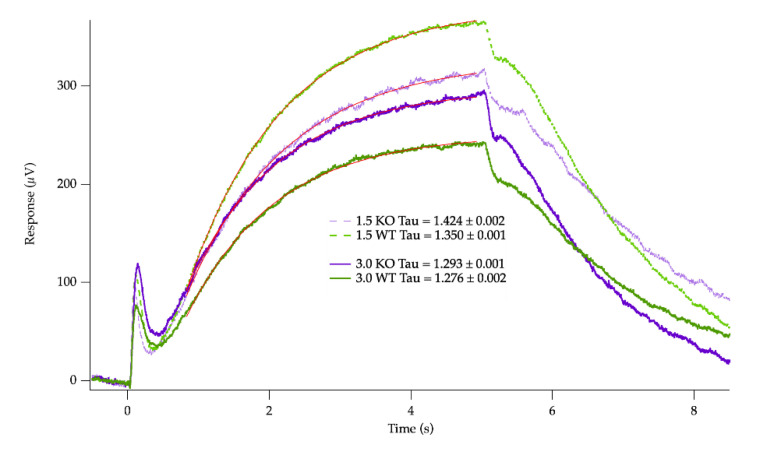
The c-wave responses to 5 s steps of light recorded from WT (green traces) or IRBP KO mice (purple traces). The dims steps (bold solid lines) were in response to 1530 R*/rod/s (3.0), whereas the bright step responses were to stimuli of 42,300 R*/rod/s (thin dashed lines, 1.5). The thin red lines indicate exponential curves fit to the data from 0.8 to 4.95 s s.stimuli (1.46 ± 0.08 s, KO: 1.40 ± 0.11 s; *p* = 0.42). The IRBP KO mice had c-waves (Figure 5 purple traces) that were very similar to each despite the 28-fold increase in stimulus intensity, while the WT mice showed strikingly different c-wave amplitudes (Figure 5 green traces). A significant difference was seen when the ratios of the dim vs. bright light response c-wave amplitudes were calculated by the higher intensity response and comparing that ratio between groups (WT: 1.59 ± 0.09, KO: 1.13 ± 0.04, *p* = 0.0004).

**Figure 6 ijms-24-10655-f006:**
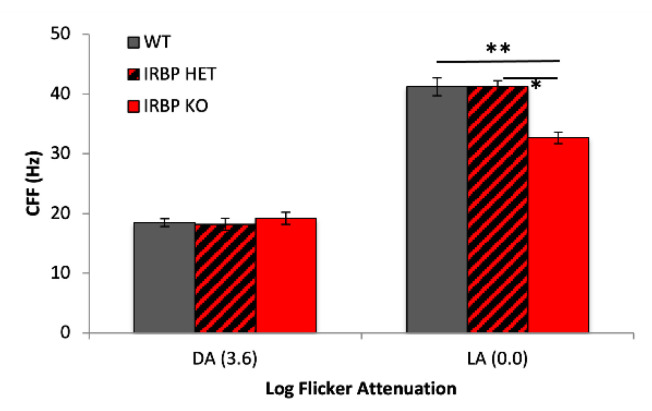
Photopic flicker sweep ERG responses were used to calculate critical flicker frequency. IRBP KO mice show reduced CFF under photopic (0.0 or 1.2 log units attenuation) (red bars) compared with WT (gray bars) or IRBP Het animals (red/black-hatched bars). Stimulus frequencies swept from 0.1 to 30 Hz for scotopic flicker and 0.1–55 Hz for photopic flicker (WT mice *n* = 12, IRBP KO mice *n* = 13, IRBP Het mice *n* = 5). Analyses detailed in methods and Appendix A. * *p* < 0.05, ** *p* < 0.005.

**Figure 7 ijms-24-10655-f007:**
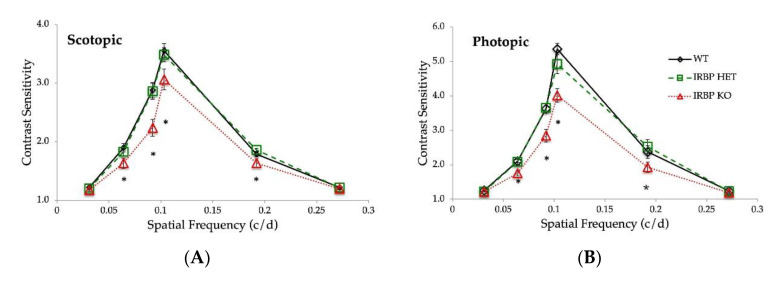
Spatial acuity and contrast sensitivity measured using optokinetic reflex (OKR). (**A**) Photopic and (**B**) Scotopic contrast sensitivity was reduced in IRBP KO mice. (**C**) Photopic and scotopic spatial acuity were also reduced in IRBP KO mice. Note: A and B are displayed on a different y scale. (* *p* < 0.05) (WT *n* = 6, IRBP KO *n* = 5, IRBP Het *n* = 7).

**Table 1 ijms-24-10655-t001:** ERG Data Summary for 1-month-old Mice.

Genotype	a-Wave (µV)	b-Wave (µV)	a-Wave ttp (ms)	b-Wave ttp (ms)	a/b Ratio (%)	CFF (Hz)	OP Area (µV*ms)	I ½ (R* Per Rod)
Dark-adapted								
WT	434 ± 39	819 ± 58	9.7 ± 0.8	67.8 ± 2.6	52.1 ± 1.7	18.5 ± 0.7	3.98 ± 0.23	21.9 ± 0.11
IRBP Het	420 ± 41	821 ± 86	8.0 ± 0.3	73.4 ± 3.4	51.6 ± 1.7	18.2 ± 1.0	3.97 ± 0.38	13.8 ± 0.16
IRBP KO	195 ± 17 **††	616 ± 58 *	11.0 ± 0.9 †	93.4 ± 3.3 **††	32.0 ± 0.9 **††	19.2 ± 0.9	2.19 ± 0.24 *†	15.6 ± 0.03
Light-adapted								
WT	252 ± 31	384 ± 46	10.6 ± 0.6	53.9 ± 3.7	67.9 ± 42	41.2 ± 1.5	2.64 ± 0.21	--
IRBP Het	196 ± 26	291 ± 31	10.6 ± 0.5	50.1 ± 1.6	66.3 ± 2.2	41.2 ± 2.6	2.26 ± 017	--
IRBP KO	109 ± 13 **†	233 ± 28 *	12.5 ± 0.7	62.8 ± 0.1	49.4 ± 1.3 **††	32.7 ± 2.1 *†	1.89 ± 0.19 *	--

vs WT * *p* < 0.05, ** *p* < 0.005; vs. IRBP Het † *p* < 0.05, †† *p* < 0.005; ttp = time to peak, OP = oscillatory potentials, CFF = critical flicker frequency. All values are mean ± s.e.m.; WT mice *n* = 12, IRBP Het mice *n* = 5–7, IRBP KO mice *n* = 11–13.

## Data Availability

Raw data are available upon request. Data are not publicly available due to the complexity of data format and storage.

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
