# Peer review of "Retinal Responses to Visual Stimuli in Interphotoreceptor Retinoid Binding-Protein Knock-Out Mice"

_ijms, 2023, doi:10.3390/ijms241310655_

Round 1
Reviewer 1 Report
Overview and general comment:
This manuscript aims to elucidate the functional role of interphotoreceptor retinoid-binding protein (IRBP) in the retina by evaluating various ERG parameters in IRBP knockout (KO) mice and linking them to retinal structural changes. In addition, the authors demonstrate adaptive potentiation (AP) in the isolated retina and verify changes in retinal pigment epithelium (RPE) function caused by IRBP deficits using c-wave measurement and analysis. The authors also use optokinetic reflex (OKR) to investigate the special frequency acuity of IRBP KO mice in both scotopic and photopic conditions.
While this study advances previous research on the functional role of IRBP and the ERG outcomes, especially the c-wave, are robust and well-verified, there are numerous essential content and formatting requirements that have been neglected. The evidence presented in the abstract and results section is somewhat anecdotal and lacks depth. There is also a lack of consistency in the sizes of important labels and indicators, which combined with small figures, makes it difficult to read and understand the information presented. Additionally, the authors should have provided a more comprehensive and connected presentation of each figure to explain and prove their goals.
For example, the presentation of figures G and H in Figure 1 is unclear, and the authors should have included them in the figure. It is also unclear why the authors measured and analyzed the scotopic threshold response shown in Figure 3C and D, as they should have focused on the results of oscillatory potential (OP) in the manuscript to interpret the inner retinal function in the IRBP deficit mice retina. Overall, the manuscript requires careful review and revised formatting to increase its readability and plausibility.
Introduction, Results, and Figures
1. The last paragraph in Introduction “Here, we report visual responses to complex stimuli, … Finally, the role of IRBP in a recently described adaptive potentiation (AP) was examined and results were consistent with an important role of IRBP in AP as well.”
It reports visual responses to complex stimuli and examines the role of IRBP in adaptive potentiation (AP), consistent with an important role of IRBP in AP. However, the authors tend to describe the experiments performed anecdotally. It would be more logical to first present and explain OCT and histological measurements, as the authors showed the structural features in IRBP KO mice before attempting to reveal functional changes and characteristics. To present the results with consistent logic in accordance with the purpose of the experiment, it is recommended that the authors restructure the presentation of their findings.
2. For all Figures
The authors have presented comparisons between IRBP Het and WT results inconsistently throughout the manuscript. While some figures, such as Figure 1, Figure 6, and Figure 7, show both IRBP Het and WT results, others do not present IRBP Het results. It is unclear why this inconsistency exists, and it would be helpful if the authors could provide a clear rationale for why some figures show both groups, while others do not. To improve the manuscript's clarity and consistency, the authors should ensure that they present the results for both IRBP Het and WT groups in a consistent manner throughout the manuscript.
3. Fig. 1,
The arrangement of the Figures in the manuscript is inappropriate, as the letter "C" comes before "A" and "B." Furthermore, it appears that Figures G and H are omitted. Additionally, there are issues with the resolution of the OCT image, where the choroidal region looks too bright in Figure B (IRBP Het) compared to Figures A and C. To improve the manuscript's quality, the authors should correct the arrangement of the Figures, include the missing Figures, and adjust the brightness of the choroidal region (Fig. 1B) to ensure consistency and clarity throughout the manuscript.
4. 2.3. Electroretinogram (ERG) in Results and Fig. 3
In Figure 2C and 2D, the authors presented ERG recordings performed on IRBP KO mice, which confirmed changes in outer retinal functions (ERG a, b waves) and inner retinal functions (scotopic threshold response (STR), OP) in IRBP KO mice. However, there are no explanations or implications for the results of Figure 2C and 2D anywhere in the manuscript. Furthermore, the authors did not provide detailed detection conditions for the STR in the method section. Even under optimal conditions, the pSTR comes out very small (around 20 uV), and the filtering range (1.6~300 Hz) provided by the authors in the Figure legend is not a STR-specific condition. This waveform appears to be a more reproducible rod-driven b-wave at the threshold level just before the scotopic a-wave start to occur, rather than a typical STR. To improve the manuscript's quality, the authors should include detailed explanations and implications for the results presented in figure 2C and 2D and provide more information on the detection conditions for the STR in the method section.
5. Fig 3E & D
The changes in OPs under scotopic and photopic conditions are not presented quantitatively, which makes it difficult to fully grasp the extent of the changes. It would be beneficial to include a quantitative graph to better illustrate these differences.
6. Fig. 7
The labels A, B, and C are missing in the Figure file and the array overlaps, causing confusion between the label and sub-title (special acuity or special frequency). It is recommended to review and revise the figure carefully to ensure clarity and accuracy.
Reference citation
Many errors have been identified in the referencing of sources. It is recommended to carefully proofread and correct the errors. A few examples are provided below for reference.
Ex.1 In the last paragraph of Page 5 (Results),
“To determine if IRBP plays a key role in adaptive potentiation (AP) previously reported by our lab (McKeown and Kraft, 2014; McKeown and Kraft, 2016), massed PR response recordings were made in isolated tissues with synaptic blockers.”
The reference for "McKeown and Kraft, 2016" is missing in their reference list. Please review and update the reference list accordingly.
Ex. 2 In the first paragraph of Page 8 (Discussion)
These general findings are in agreement with the original descriptions of this line of mice (Markand et al. 2016; Ripps et al. 2000; Ripps et al. 2000; Wisard et al. 2011; Wisard et al. 2011; Sato et al. 2013; Sato et al. 2013).
The manuscript contains instances where the same references are cited unnecessarily and repeatedly, indicating the need for careful review and correction to avoid such citation errors.
Ex. 3 In the first paragraph of Page 9 (Discussion)
"These results suggest that the cone circuits are more sensitive to PR loss and agree with a similar finding in light-damaged albino rats with modest losses of cone PRs (Benthal et al. 2022; Rubin et al. 2022)".
The reference for "Benthal et al. 2022" is missing in their reference list. Please review and update the reference list accordingly.
Ex. 3 In the third paragraph of Page 9 (Discussion)
" All-trans retinol is thought to be part of a feedback loop, resulting in AP by increasing sensitivity of rods (McKeown and Kraft, 2014; McKeown and Kraft, 2016)".
The reference for "McKeown and Kraft, 2016" is missing in their reference list. Please review and update the reference list accordingly.
Discussion & Method
1. In the second paragraph of Page 8
“The OP area was reduced by 44% under dark-adapted conditions, and by 17% under light-adapted conditions, suggesting a significant loss in cone driven signals in bipolar cell to amacrine and amacrine to ganglion cell signaling in the IPL”
The authors have drawn a conclusion regarding the significant loss in cone-driven signals from bipolar cells to amacrine and amacrine to ganglion cells in the IPL, based only on the OP results. However, they also measured the STR, which was mentioned earlier as a crucial problem, and found significantly reduced responses in IRBP deficient mice under scotopic conditions. The inconsistency between these findings raises questions that need to be addressed.
2. Scotopic CFF is normal despite thinning of ONL and reduced scotopic ERG in IRBP KO mice
The authors' suggestion of a redundancy or plasticity in circuits that detect (encode) scotopic flicker at threshold, even with substantial rod loss, should be supported with additional evidence.
3. Overall, a more in-depth supplementary interpretation of the following significant findings should be needed.
The authors obtained a wide range of technically challenging ERG parameters, but the co-relative interpretation of each outcome is weak, and a more detailed analysis and interpretation of the relationship between inner retinal function (with their OP and STR results) evaluation and OKR is lacking. For instance, they should explain the discrepancy between the Op results (which indicate that photopic vision is more vulnerable than scotopic vision) and the OKR results (which show no significant difference between both conditions) in the discussion.
Nothing to mention.
Author Response
Please see the attachment for responses to both reviewers comments.

Reviewer 2 Report
The authors of the manuscript investigated interphotoreceptor retinoid binding protein (IRBP) knock out mice using a combination of histological, electrophysiological and behavioral methods. IRBP knock out mice have been studied before by other groups and several findings shown in the manuscript had been reported before. This includes reduction in ONL thickness, loss of photoreceptor cells, reduction in the photopic and scotopic ERG amplitude, reduction in photopic CFF. Hence the manuscript provides little new principle findings. However, the authors carried out a more detailed electrophysiological characterization of IRBP knock out mice and also an optokinetic reflex assay. An interesting conclusion of their results is that retinas lacking IRBP might develop a compensatory mechanism resulting in higher synaptic gain between PRs and bipolar cells, although the mechanistic basis for this effect remains unclear. The experiments were meticulously conducted and documented and the manuscript was well written. I only have a few minor suggestions for improvement.
1) Het mice presumably are mice heterocygous for IRBP mutation. This should be clarified in the manuscript
2) Please also explain what the target of mCarr is and why it specifically labels cones.
3) Figure 1: panels (G) and (H) mentioned in the figure caption are missing (data are included in D and F).
Author Response

(The authors gave the same response as above.)
